# Effect of Fly Ash on Leaching Characteristics of Cement-Stabilized Macadam Base

**DOI:** 10.3390/ma14205935

**Published:** 2021-10-09

**Authors:** Dian Lun, Taiping Yuan, Xiaolong Yang, Hongliu Rong, Junjie Shi, Minqiang Pan

**Affiliations:** 1College of Civil Engineering and Architecture, Guangxi University, Nanning 530004, China; lundian2020@126.com (D.L.); ronghongliu@126.com (H.R.); shijunjie2021@126.com (J.S.); minqiang73@163.com (M.P.); 2Energy Development Research Institute Co., Ltd., China Southern Power Grid, Guangzhou 510530, China; yuantp@csg.cn

**Keywords:** cement-stabilized macadam base, leaching, fly ash, cement dosage

## Abstract

To study the leaching characteristics of a cement-stabilized macadam base with fly ash, a calcium leaching test, using varying cement and fly ash dosages in an ammonium chloride solution, was designed to obtain the rate of calcium ion leaching, porosity, and permeability coefficient of cement-stabilized macadam with leaching time. The results showed that the number of leached calcium ions increased with the cement dosage in the leaching of cement-stabilized macadam. With an increase in the cement dosage, the leaching rate of calcium ions decreased, and the leaching process of the material was delayed. The incorporation of fly ash could effectively slow down the degree of calcium ion leaching. The permeability coefficient increased gradually with the extension of leaching time, and the increase in fly ash content had a more significant effect on the improvement of the permeability coefficient than the increase in cement dosage did.

## 1. Introduction

Cement-stabilized macadam base, exhibiting good stability, strength, and low cost, has been widely used in China’s highway construction. Cracks will inevitably occur in the pavement base, and pavement seepage and groundwater can enter the pavement base owing to its construction. However, the drainage performance of the pavement base is poor. Some of the water is retained between the surface and base, causing scouring damage to the pavement base, which is under a frequent load. Some of this seepage enters the pores in the base material, extracting calcium ions, resulting in road surface mud and potentially severe damage. Water damage is the main form of damage to cement-stabilized macadam bases. Existing research has focused mainly on scouring damage, and there is a lack of research on leaching loss.

Currently, some scholars at home and abroad have studied leaching problems. For example, Guo found that the axial compressive strength, splitting tensile strength, and quality of concrete gradually decreased with age owing to erosion. Guo also found that the degradation of the splitting tensile strength was more evident than that of the axial compressive strength [1]. Zhou et al. found that the elastic modulus, compressive strength, and splitting tensile strength of concrete decreased with increased leaching time [2]. Kong et al. found that leakage dissolution leads to a decrease in the mechanical properties of concrete and a significant decrease in the splitting tensile strength [3]. In a soft water dissolution environment, Guo found that the Ca^2+^ in the cement paste continued to leach, and the limit value of the cumulative leaching ratio of Ca(OH)_2_ in the sample was approximately 30–40% of the total. The number of years taken for the compressive strength of the cement slurry sample to decrease significantly was closely related to the water-binder ratio [4]. Deng et al. studied the law of calcium ion leaching in different cycles by measuring the calcium ion content in a corrosive solution [5,6,7]. Faucon found that a significant amount of calcium and a small amount of silicon were leached in the surface leaching area, and that the leaching of cement-based materials is an inherent problem. The local chemical equilibrium controls the leaching process of concrete in soft water, and the calcium leaching process is controlled by the diffusion of calcium ions [8,9]. Haga conducted an experimental study on the leaching of cement stone with varying water–cement ratios. The results showed that the porosity increased continuously with the leaching of calcium ions, and that the increase in porosity accelerated the diffusion and leaching of calcium ions [10]. Tognazzi established an empirical relationship between the effective diffusion coefficient of calcium ions and porosity [11]. Bilal’s study showed that the morphological changes caused by calcium leaching indicated that the solid volume fraction in the cement matrix decreased, thereby increasing the porosity, and ultimately, reducing the strength and durability of permeable concrete [12]. Song et al. studied the leaching behavior of calcium in one dimension (the other four surfaces of the test sample were sealed, and the leaching rates of calcium ions from the upper surface to the lower surface of the test sample were analyzed) in a cement slurry by immersing the samples in an NH_4_Cl solution. They found that the electrical resistance in the leaching zone increased, which can be used as a measure to characterize the degree of leaching [13]. Amenta et al. accelerated the leaching at a voltage of 30 V. Their SEM results confirmed the migration of Ca^2+^ and the self-repair mechanism of newly formed calcite precipitated in the microstructure of cement-based materials [14]. Long et al. studied the leaching resistance of graphene oxide using the deionized water leaching method, and considered that the oxygen-containing functional groups of graphene oxide could capture Ca^2+^ and reduce the leaching of calcium [15].

According to research and surveys, approximately 1.3 billion tons of fly ash waste is produced globally every year; however, its recycling rate is low, and approximately 47% of the fly ash waste goes to landfills. The existence of several fly ash landfills will negatively affect the ecological environment and human health [16]. Fly ash is a pozzolanic active material. Its main chemical components are SiO_2_ and Al_2_O_3_, and its mineral is mainly an aluminum-silicon vitrified structure. The application of fly ash in cement-based materials can improve the strength and durability of these materials, and may mitigate the harm caused by fly ash to the environment. The primary mechanism of action of fly ash in cement-based materials is based on its physical activity: particle, micro-aggregate, and compactness effect [17]. Ferdous et al. found that adding fillers, such as fly ash, to the resin improves the heat resistance and durability of the epoxy matrix [18]. Yu et al. mixed epoxy resin with a flame-retardant filler, fly ash, and other functional fillers to improve the durability [19]. Fang studied the pore structure and leaching resistance of RCC (roller-compacted concrete) with low cement and high fly ash content [20]. Roziere’s research showed that the addition of fly ash reduced the calcium hydroxide content in the concrete and decreased the soluble calcium hydroxide content in the material, thereby decreasing the degree of leaching damage [21]. Roziere’s research also showed that fly ash alone can slow down the calcium leaching process of cement paste and improve its corrosion resistance in deionized water [22]. Through microscopic analysis, Sun found that, when the fly ash content was not more than 50%, the calcium silicate hydrate (CSH) content increased continuously, and the number of nondetrimental pores increased significantly [6]. In Sun’s investigation, it was found that the addition of fly ash could reduce the porosity. However, Fang’s test results showed that, although the addition of fly ash reduces the calcium hydroxide (CH) content and the number of calcium ions leached out, it increases the porosity of the sample, and accelerates the calcium leaching process of the material [23].

Existing research on the leaching and scouring of cement-stabilized macadam base has mainly focused on the thermodynamics of dissolution and precipitation reactions of the calcium hydroxide produced by the hydration of cement and the decomposition reaction kinetics of calcium silicate hydrate. Other aspects include the migration mechanism of corrosion reaction products in concrete, micro-characterization of pore structure, and porosity changes caused by corrosion. However, it is challenging to achieve dissolution equilibrium under the action of ambient water, and the change in the base material properties caused by dissolution has not been studied thoroughly. Moreover, there are few studies on the influence of fly ash on the dissolution properties of cement-stabilized macadam materials.

During the leaching process of cement-based materials, the solution in the pores continuously diffuses out to dissolve the calcium hydroxide, resulting in a decrease in the pH of the pore solution. Concurrently, the dissolution of solid-phase-bound calcium also produces more pores in the cement-based materials and increases the connectivity of the interior pores. Therefore, in the research on calcium leaching of cement-based materials, the quantity of leached calcium ions and the change in porosity are commonly used to characterize the calcium leaching process [3,5,6,7,8,9,10,11].

Therefore, to study the leaching characteristics of cement-stabilized macadam with fly ash, an accelerated leaching test of cement and fly ash with varying additions of ammonium chloride solution was designed. The variations in calcium ion leaching, porosity, and permeability of cement-stabilized macadam were obtained through tests, and the results were analyzed accordingly.

## 2. Materials and Methods

### 2.1. Materials

#### 2.1.1. Cement

In this study, P×O42.5 Portland cement was used. According to the Methods for Chemical Analysis of Cement (GB/T 176-2017), the chemical composition is shown in Table 1, and the mineral compositions are shown in Table 2.

#### 2.1.2. Fly Ash

Grade I low-calcium fly ash was used, and its chemical composition is listed in Table 3.

#### 2.1.3. Aggregate

The aggregates used herein were gravel and river sand. The fraction with a particle size greater than 4.75 mm was used as the coarse aggregate. The fine aggregate was river sand. The crushing value of the gravel material was 20.6%, and the water content of the aggregate was 0.3%. The apparent density of the fine aggregate was 2.58 g/cm^3^. After examination, each index of the aggregate met the requirements of the specifications.

#### 2.1.4. Water for Test

The cement-stabilized macadam was prepared with potable water. Moreover, 6 mol/L of ammonium chloride solution and EDTA (Ethylene Diamine Tetraacetic Acid) standard solution were prepared with deionized water (pH 7.0).

#### 2.1.5. Ammonium Chloride

Ammonium chloride solution tests were prepared using ammonium chloride powder. The ammonium chloride content was not less than 99.5%, and the relative molecular mass was 53.49. The solubility of the ammonium chloride in water at room temperature is approximately 37.2 g/100 mL.

#### 2.1.6. EDTA Standard Titration Solution

According to the Test Methods of Materials Stabilized with Inorganic Binders for Highway Engineering (JTG E51-2009), analytical-reagent-grade disodium EDTA was used to produce the standard titration solutions of EDTA at 10 and 100 mmol/L. In addition, 1:1 diluted hydrochloric acid and 20% sodium hydroxide solution, CMP indicator, E44 epoxy resin were used.

### 2.2. Mixture Design

#### 2.2.1. Gradation Design

The results of the grain-gradation calculations at differing remaining voids are shown in Table 4.

The cement/fly ash ratios mixed into the macadam were 1:0.1, 1:0.2, and 1:0.3, which were compared with samples without fly ash.

#### 2.2.2. Optimal Water Content and Maximum Dry Density

The water content–dry density curve of the mixture was drawn from the results obtained using the heavy compaction method, according to the Test Methods of Materials Stabilized with Inorganic Binders for Highway Engineering (JTG E51-2009). This allowed the optimum water content and maximum dry density of the sample to be determined. Samples with five different water contents were prepared for each fly ash level. The test results are presented in Table 5.

#### 2.2.3. Preparation and Maintenance of Samples

All test pieces were shaped using static pressure. After demolding, the samples were sealed in plastic bags and held for 90 days in standard maintenance rooms at a temperature of 20 ± 2 °C and a relative humidity of more than 95%. According to the results in Table 5, three parallel specimens were made for each variable. The samples obtained after this process were sealed with epoxy resin to ensure that they were only leached in the axial direction. Each sample was cut into two halves, half of which was taken to be the test block. As shown in Figure 1, these are the specimens prepared in the experiment.

### 2.3. Experimental Methods

#### 2.3.1. Accelerated Leaching of Ammonium Chloride

Gérard performed an accelerated concrete leaching test with a 6 mol/L NH_4_NO_3_ solution, proposed a 1D simulation model of calcium leaching of concrete, and verified the model’s reliability [24]. Subsequently, many scholars globally have used this method to study the mechanism of calcium leaching [2,5,10,25,26,27]. Research shows that little difference exists between accelerated leaching with ammonium chloride solution and that under a natural environment [10]. Therefore, a 6 mol/L ammonium chloride solution was used to accelerate the leaching test.

The ammonium chloride-accelerated leaching method was used to simulate the leaching process. First, a 6 mol/L ammonium chloride solution was prepared. The sample was then placed in the ammonium chloride solution for dissolution corrosion; it was ensured that the upper and lower surfaces were in complete contact with the solution. Finally, the container was sealed during leaching. During the leaching process, the calcium ion concentration in the corrosion solution was measured every two days. The porosity and permeability coefficient of the sample were measured at intervals of four and eight days. The total test period was 28 days.

#### 2.3.2. Calcium Ion Concentration Determination

After the corrosion solution was thoroughly mixed, 5 mL of solution was taken and diluted X times with deionized water. Next, 50 mL of aliquot was added to an Erlenmeyer flask. Then, three drops of dilute hydrochloric acid (1:1) were added. The solution was then boiled to discharge the carbon dioxide. Then, 5 mL of 20% sodium hydroxide solution and 80 mg of the calcein indicator were added to the flask and titrated against the EDTA solution. The solution was titrated with the configured EDTA solution. The initial volume, *V*_1_, was recorded. The titration endpoint is represented by a change from yellow/green to orange/red, as shown in Figure 2. The final volume, *V*_2_, was recorded. The calcium ion concentration was calculated as follows:(1)$$C=\frac{\left(V_{2}-V_{1}\right)\times M\times X}{V_{W}}$$
where $V_{1},V_{2}$ are the volumes of the EDTA solution before and after the test (mL), respectively; M is the EDTA concentration (mmol/L); X is the dilution factor of the solution to be tested; $V_{W}$ is the test solution volume (mL).

#### 2.3.3. Porosity Test

According to the ASTM C1202, we experimented with the saturated water-dry weighing method. First, the leached sample was removed from the solution. Next, the excess water on the surface was wiped off with a towel and weighed. Then, the sample was completely immersed in water and weighed using a hanging scale. Subsequently, the sample was placed in an oven at 60 °C and dried to a constant weight, which was recorded. Finally, the porosity of the sample was calculated as:(2)$$\varphi =\frac{m_{s}-m_{d}}{m_{s}-m_{x}}\times 100\%$$
where $m_{s}$ is the saturated mass of the sample (g); $m_{x}$ is the mass of the sample immersed in water (g); $m_{d}$ is the mass of the sample after drying (g).

#### 2.3.4. Water Permeability

Using the methods for inorganic binders as stabilizing materials in highway engineering (JTG E51-2009), the diameter of the base of the seepage meter was set to 100 mm, and the permeability coefficient of the sample was measured by changing the head. First, the sample was padded to ensure that the seepage was unhindered. Then, the bottom of the instrument was placed on top of the test sample, and the contact area between the instrument and sample was sealed to prevent water seepage from affecting the test results. Next, the instrument was filled with water to a suitable depth, and the lower switch was opened quickly. When the water level dropped to *V*_1_ = 100 mL, the recording time started. As the descent process was slow, only a 3 min seepage volume was used to calculate the permeability coefficient. The volume *V*_2_ in the seepage pipe was recorded at this time. Finally, the permeability coefficient was calculated as
(3)$$C_{W}=\frac{V_{2}-V_{1}}{t}$$
where $\;V_{1},V_{2}$ are the volumes of water in the container before and after the test (mL), respectively; t is the time taken for testing (min).

If the sample was not corroded, Ca(OH)_2_ in the test piece turned magenta when exposed to phenolphthalein. If the test piece was leached, the calcium ions in the sample would leach out, and the sample would not change color when exposed to phenolphthalein. Using this distinction, the degree of corrosion of the sample was estimated.Figure 3 shows that, over time, the color development of phenolphthalein gradually decreased. That is, the degree of leaching of the sample decreased, and the calcium ions were almost completely leached. 

## 3. Results and Discussion

### 3.1. Leaching Amount of Calcium Ion

#### 3.1.1. Effect of Cement Dosage on Calcium Ion Concentration

Figure 4a–d show the calcium ion concentration change in the ammonium chloride corrosion solution over time with different cement dosages and fly ash contents. When the amount of fly ash was fixed, the calcium ion concentration increased with the cement dosage. This was because the pore calcium ions in contact with the solution in the sample reacted with ammonium chloride, disrupting the dissolution balance of calcium ions in the pore. Consequently, the solid-phase calcium ions continuously dissolved into the pore solution. This resulted in a difference in the concentration of calcium ions in the pores of the solution. Thus, the calcium ions in the pores continued to disperse into the solution. Finally, the calcium ion concentration in the NH_4_Cl solution increased continuously. Increased cement dosage increased the contact area between the cement hydration products and corrosive fluid. Therefore, the probability of leaching of calcium ions by corrosion increased, which ultimately led to an increase in the content of calcium ions in the corrosion solution.

Figure 4a–d show that, when the cement dosage was 7% and no fly ash was added, the concentration of calcium ions leached out at the early stage of leaching was significantly greater than those at other cement dosages. However, with the increase in fly ash content, the gap between the two was significantly reduced. The former was because, when the cement dosage reached 7%, some cement hydration products were not involved in the stability of the gravel, but remained on the outside of the sample. Consequently, a considerable amount of calcium hydroxide reacted with ammonium chloride at the early stage of leaching, and the concentration of calcium ions in the corrosive solution was higher than those in other cement dosages. The latter was because a considerable amount of calcium hydroxide was converted into hydrated calcium silicate by the addition of fly ash, and the dissolution rate of hydrated calcium silicate was reduced. Therefore, the quantity of calcium ions leached out was small in the early stage of leaching.

#### 3.1.2. Effect of Cement Dosage on the Calcium Ion Leaching Rate

According to Figure 5a–d, when the amount of fly ash was constant, the leaching rate of calcium ions decreased gradually with the cement dosage. This was because an increase in the cement and fly ash dosage reduced the porosity of the sample. However, the leaching process of the material started from the outside and progressed inside. When the porosity decreases, it is more difficult for the corrosive solution to enter the pores of the material, and the calcium ion leaching rate (the ratio of leached calcium ions to the calcium content of CH and CSH in the specimen) decreases.

Figure 5a shows that, when fly ash was not added, the calcium ion leaching rates of the 3% and 4% cement samples were significantly higher than those of the other cement samples. Figure 5b–d show that, when fly ash was added, the leaching rate of the calcium ions at a cement dosage of 4% was significantly improved. When the cement dosage was 3%, the leaching rate of calcium ions decreased, although it was still higher than those at other cement dosages. When the cement dosage was 3%, the leaching rate of the calcium ions decreased; however, it was still larger than those at other cement dosages. Therefore, the leaching damage caused by calcium ion leaching can be mitigated by increasing the cement dosage. When the base material is not mixed with fly ash, a 5% cement dosage should be considered as the lowest dosage. When the fly ash is mixed in the base material, a 4% cement dosage should be considered as the lowest dosage.

#### 3.1.3. Effect of Fly Ash Content on Calcium Ion Concentration

According to the analysis of Figure 6a–e, when the cement dosage was fixed, the amount of calcium ion leaching decreased gradually with fly ash content. However, the quantity of calcium ions leached out under different cement dosages and fly ash contents was slightly different. When the cement dosage was 3%, the addition of fly ash did not significantly reduce the leaching of calcium ions. When the cement dosage was more than 3%, the slowing effect of the fly ash on calcium ion release increased with cement dosage. This is because, when the content of fly ash is relatively small, a significant amount of active silica in the fly ash reacts fully with the cement hydration products, which significantly reduces the content of calcium hydroxide in the samples and increases the content of hydrated calcium silicate [28,29]. In addition, during the leaching process, hydrated calcium silicate is more difficult to dissolve than calcium hydroxide, which results in a decrease in the amount and rate of calcium ion dissolution. Others believe that the leaching of calcium ions is also related to the calcium–silicon ratio in the mixture [30]. When the molar ratio of CaO/SiO_2_ in the mixture was greater than 1, CaO could easily be leached. When the molar ratio of CaO/SiO_2_ was less than 1, SiO_2_ leaching occurred more easily. When the molar ratio of CaO/SiO_2_ in the mixture was close to 1, the leaching of both was negligible. The addition of fly ash changed the ratio of CaO/SiO_2_ to make it closer to 1, thus reducing the leaching rate of the calcium ions.

### 3.2. Porosity Variation

#### 3.2.1. Effect of Cement Dosage on Porosity

Figure 7a–d show that the porosity increased with the leaching time. When the fly ash content was constant, the porosity gradually decreased with the cement dosage.

This was because the cement hydration products covered the crushed stones and filled the remaining pores in the process of sample formation. When the cement dosage increased, more hydration products filled the remaining pores, decreasing the remaining porosity. In addition, an increase in the cement dosage will inevitably lead to an increase in the hydration products. More active silica in the fly ash reacts with calcium hydroxide to form calcium silicate hydrate [28,29]. Calcium hydroxide mainly generates the pores. Hydrated calcium silicate mainly produces gel pores, which leads to a decrease in the number of pores and an increase in the number of gel pores [28,29]. However, because the number of pores is higher than that of the gel pores, the porosity of the entire sample decreases.

As shown in Figure 8, the content of fine crushed stones in the corrosive liquid bucket changed. During the test, it was found that the fine crushed stones falling off because of the loss of surface cement cementation in the corrosive solution gradually increased with time. The falling off of fine crushed stones caused the internal hydration products to contact the corrosive solution and aggravate the degree of leaching.

#### 3.2.2. Influence of Fly Ash Content on the Porosity

According to the analysis of Figure 9a–e, when the cement dosage was constant, the porosity decreased with the fly ash dosage. This is because, in the cement fly ash-stabilized macadam system, the fly ash mainly exhibits a pozzolanic and micro-aggregate effect [16]. The pozzolanic effect occurs in the early stage of the strength formation of cement-stabilized macadam. The rapid hydration of cement provides early strength and produces a significant amount of calcium hydroxide. The fly ash, containing active silica, can react with calcium hydroxide to produce hydrated calcium silicate with smaller pores, which reduces the overall porosity of the sample [29,31,32,33]. The micro-aggregate effect occurs when the fly ash is added to the cement-stabilized macadam material. The aggregate without the fly ash is not compact. The spherical particles of the fly ash can fill the micropores formed by the sand, and the aggregate compactness is significantly improved [31,32,33]. The cement hydration products are filled and dispersed between the fly ash particles or between the pores formed by the fly ash and sand. The cement distribution is more uniform, and the cement dosage can be reduced, thereby reducing the friction between the sand and stone and increasing the fluidity of the cement stone. This reduces the increase in pores caused by adding excessive water due to increasing fluidity.

### 3.3. Permeability

According to Figure 10a–d, the permeability coefficient gradually increased with the leaching time. According to Figure 7a,b and Figure 9b,c, this was mainly because the porosity increased with leaching time, leading to a gradual increase in the permeability coefficient.

When the fly ash was at a constant dosage, the increase in cement dosage had little effect on the permeability coefficient. However, when the dosage of cement was constant, the addition of fly ash improved the permeability coefficient. As shown in Figure 7a,b and Figure 9b,c, the porosity decreased somewhat with the proportion of cement, and the porosity decreased more with the proportion of fly ash.

## 4. Conclusions

An accelerated leaching test of cement-stabilized macadam material in a 6 mol/L ammonium chloride solution was developed. The leached calcium ion content in the solution with different cement and fly ash proportions was measured via EDTA titration. The porosity of the sample was determined using a water-dry weighing method. The permeability coefficient of the samples was measured using the variable water head method. Subsequently, the test results were analyzed. Finally, the following conclusions were drawn.

(1)During the leaching of cement-stabilized macadam material, the leaching of calcium ions increased with the cement dosage. For example, when the cement dosage was 7% and no fly ash was added, the quantity of calcium ions leached out at the early stage of leaching was significantly greater than those of the samples at other cement dosages. However, with the increase in the fly ash content, the gap between the two was significantly reduced.(2)With an increase in the cement dosage, the porosity of the samples and the leaching rate of the calcium ions reduced, and the leaching process was delayed. The cement dosage should be at least 5% when the base material is not mixed with the fly ash. The minimum cement dosage should be 4% when the base material is mixed with the fly ash.(3)The addition of fly ash can effectively reduce the degree of calcium leaching. However, the effect of fly ash was not the same with different cement dosages. When the cement dosage was 3%, the addition of fly ash did not significantly reduce the leaching of calcium ions. When the cement dosage was greater than 3%, the slow-down effect of the fly ash on calcium ion leaching increased continuously with the cement dosage in the test samples.(4)In the leaching process of the cement-stabilized macadam material, the permeability coefficient increased gradually with the leaching time. Therefore, the increase in fly ash content had a more significant effect on the improvement of the permeability coefficient than the increase in cement dosage did.

The test results showed that the incorporation of fly ash can reduce the extent of calcium ion leaching and porosity of the cement-stabilized macadam, reduce the material’s permeability, and improve the leaching durability and structural life of the cement-stabilized macadam base. In addition, the study of its influence on cement-stabilized macadam can improve the application of fly ash and guide engineering applications in obtaining the best mineral mixture.

## Figures and Tables

**Figure 1 materials-14-05935-f001:**
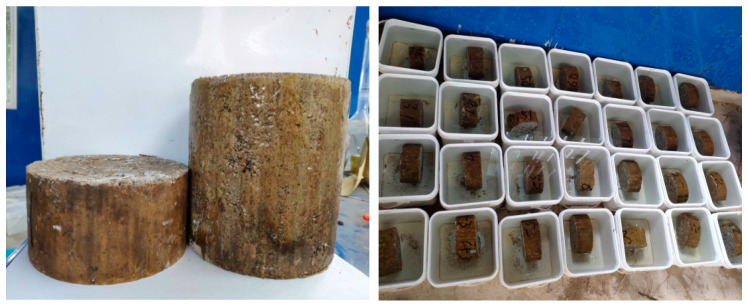
Cement-stabilized macadam samples.

**Figure 2 materials-14-05935-f002:**
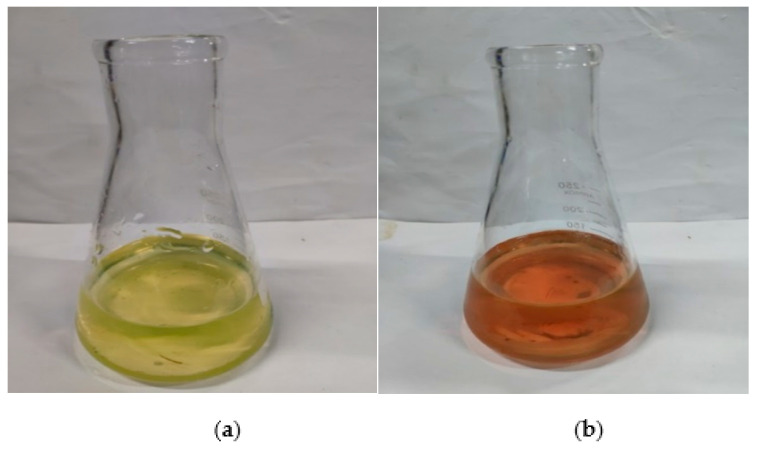
Color change of solution before and after titration: (**a**) color before titration; (**b**) color after titration.

**Figure 3 materials-14-05935-f003:**
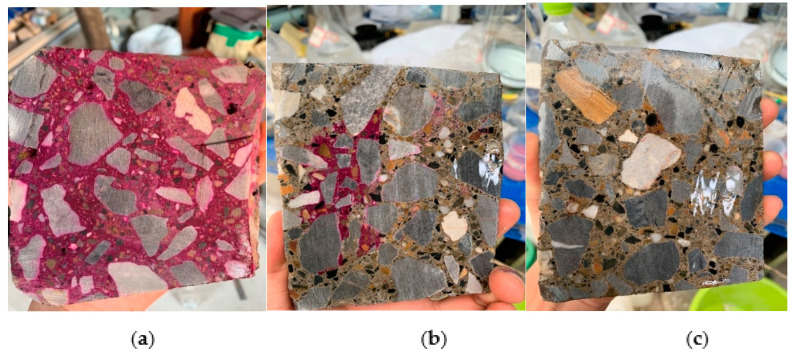
Phenolphthalein indicator shows changes in solution: (**a**) 0 h; (**b**) 1 h; (**c**) 2 h.

**Figure 4 materials-14-05935-f004:**
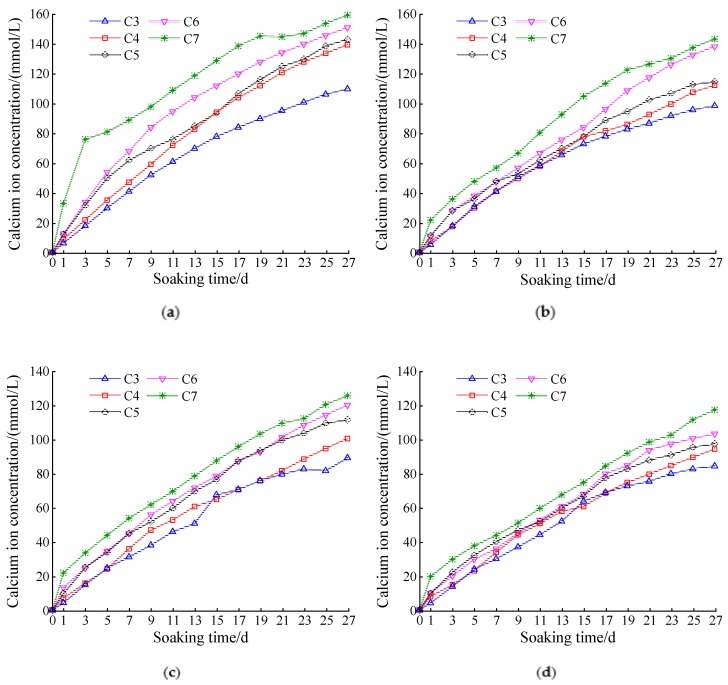
Calcium ion concentration curve at different cement dosages: (**a**) cement:fly ash = 1:0; (**b**) cement:fly ash = 1:0.1; (**c**) cement:fly ash = 1:0.2; (**d**) cement:fly ash = 1:0.3.

**Figure 5 materials-14-05935-f005:**
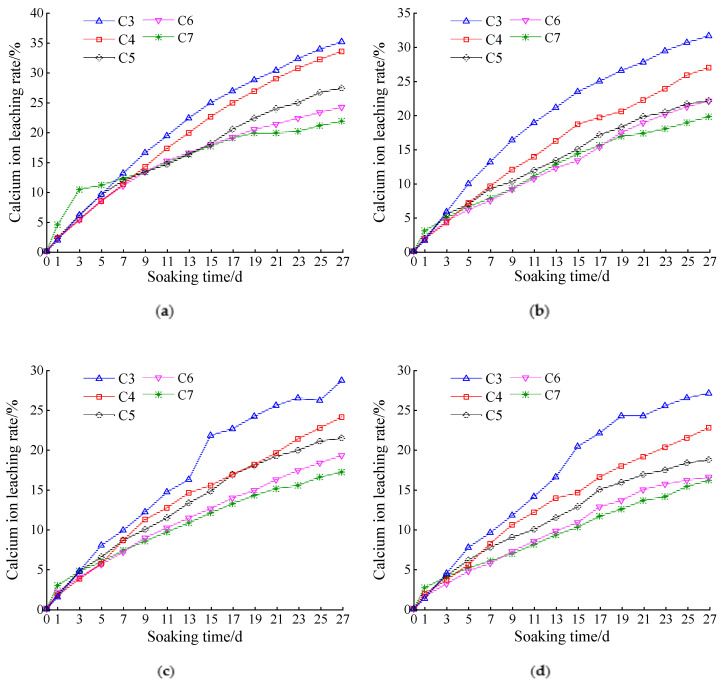
Variation curves of Ca ion dissolution rate at different cement dosages: (**a**) cement:fly ash = 1:0; (**b**) cement:fly ash = 1:0.1; (**c**) cement:fly ash = 1:0.2; (**d**) cement:fly ash = 1:0.3.

**Figure 6 materials-14-05935-f006:**
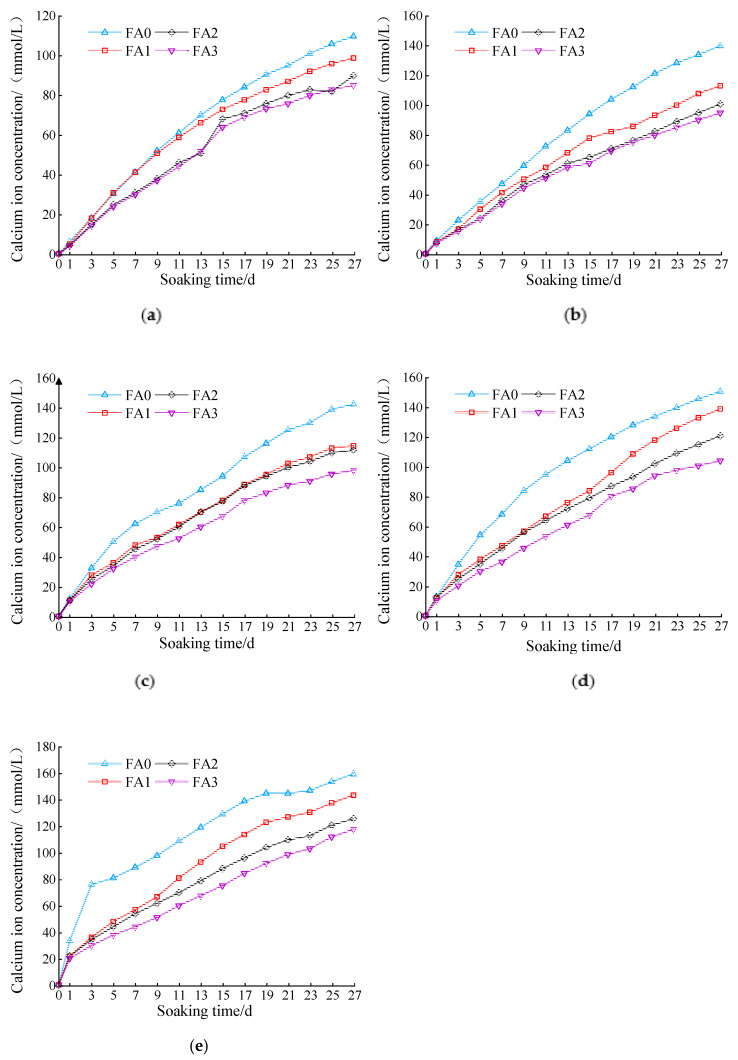
Variation curves of calcium ion concentration at different fly ash contents: (**a**) cement content: 3%; (**b**) cement content: 4%; (**c**) cement content: 5%; (**d**) cement content: 6%; (**e**) cement content: 7%.

**Figure 7 materials-14-05935-f007:**
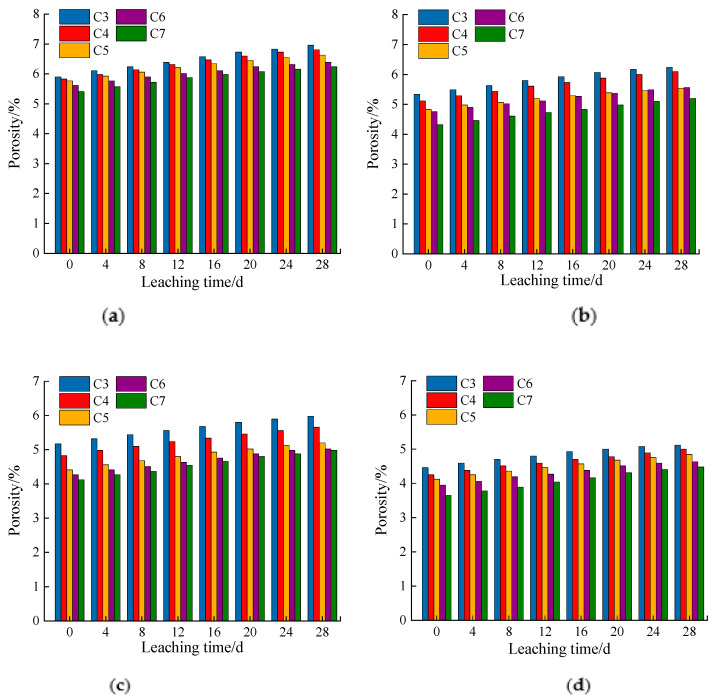
Porosity changes of samples with different cement dosages: (**a**) cement:fly ash = 1:0; (**b**) cement:fly ash = 1:0.1; (**c**) cement:fly ash = 1:0.2; (**d**) cement:fly ash = 1:0.3.

**Figure 8 materials-14-05935-f008:**
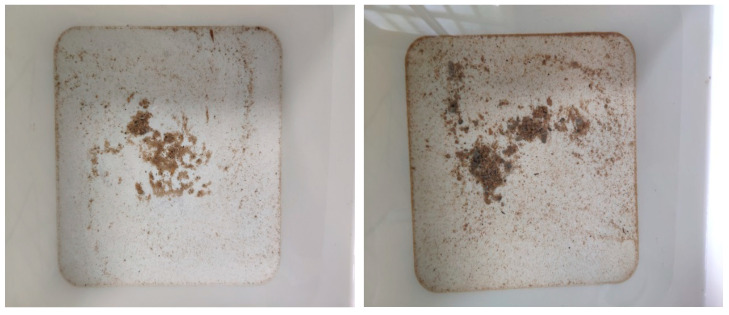
Change in fine gravel content in the corrosive liquid drum.

**Figure 9 materials-14-05935-f009:**
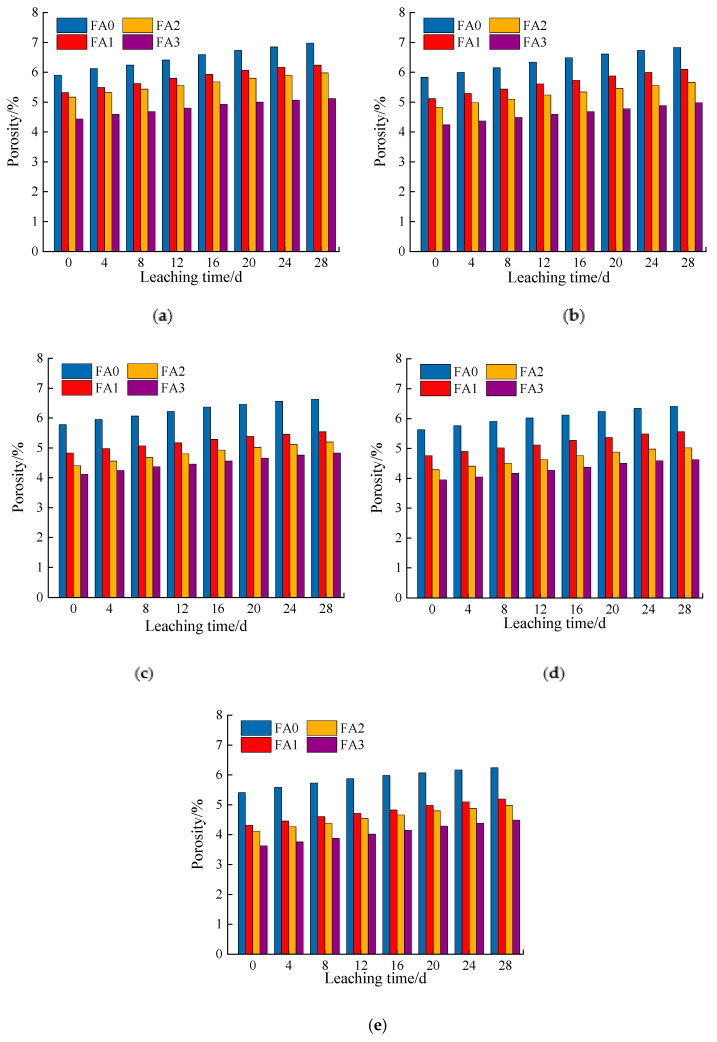
Porosity changes of samples with different fly ash contents: (**a**) cement content: 3%; (**b**) cement content: 4%; (**c**) cement content: 5%; (**d**) cement content: 6%; (**e**) cement content: 7%.

**Figure 10 materials-14-05935-f010:**
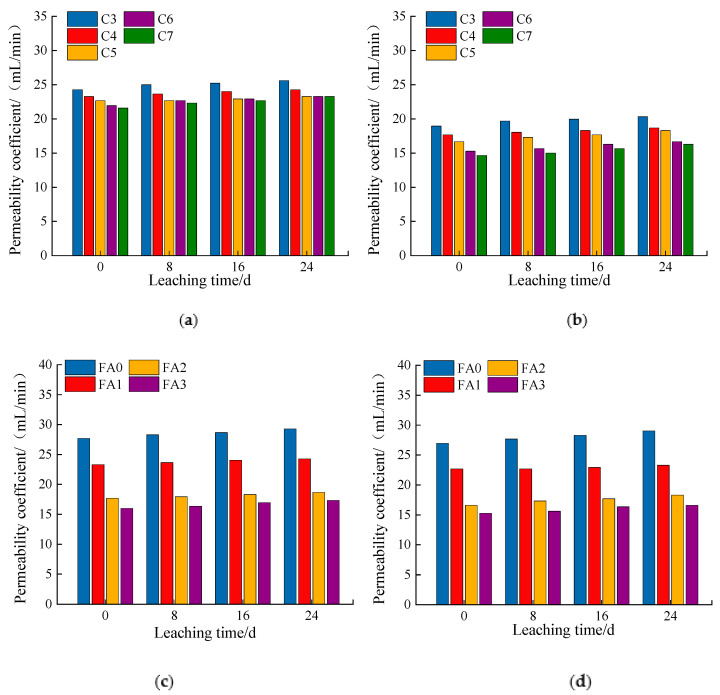
Change in permeability coefficient: (**a**) cement:fly ash = 1:0.1; (**b**) cement:fly ash = 1:0.2; (**c**) cement content: 4%; (**d**) cement content: 5%.

**Table 1 materials-14-05935-t001:** Chemical composition of cement.

Chemical Compositions	SiO_2_	Al_2_O_3_	Fe_2_O_3_	CaO	MgO	SO_3_
Content (%)	20.10	4.63	3.46	63.62	1.18	2.01

**Table 2 materials-14-05935-t002:** Mineral composition of cement.

Mineral Compositions	C_3_S	C_2_S	C_3_A	C_4_AF	C$\overset{–}{S}$H_2_
Content (%)	64.39	9.08	6.42	10.53	4.32

(Note: C_3_S—3CaO × SiO_2_, C_2_S—2CaO × SiO_2_, C_3_A—3CaO × Al_2_O_3_, C_4_AF—4CaO·xAl_2_O_3_·(1 − x)Fe_2_O_3_, C$\overset{–}{S}$H_2_—3CaSO_4_ × 2H_2_O).

**Table 3 materials-14-05935-t003:** Chemical composition of fly ash.

Components	SiO_2_	CaO	Al_2_O_3_	Fe_2_O_3_	MgO	Others
Content (%)	53.36	2.27	29.09	3.87	0.81	10.6

**Table 4 materials-14-05935-t004:** Particle size distribution design with differing residual voids.

Sieve Size (m)	Percentage of Pass at Different Residual Voids (%)				
	3%	4%	5%	6%	7%
31.5	100	100	100	100	100
19.0	67	70	72	73	74
9.5	46	49	51	54	55
4.75	31	34	37	39	41
2.36	21	24	26	29	31
1.18	14	17	19	21	23
0.6	10	12	14	15	17
0.3	7	8	10	11	13
0.15	4	6	7	8	9
0.075	3	4	5	6	7
n	5.420	5.595	5.735	5.853	5.954

**Table 5 materials-14-05935-t005:** Compaction test results of cement-stabilized macadam.

NO.	Cement Dosage	Cement: Fly Ash	Optimum Water Content	Maximum Dry Density
C3FA0	3%	1:0	4.11	2.213
C3FA1		1:0.1	4.12	2.224
C3FA2		1:0.2	4.22	2.233
C3FA3		1:0.3	4.30	2.247
C4FA0	4%	1:0	4.41	2.251
C4FA1		1:0.1	4.52	2.273
C4FA2		1:0.2	4.65	2.284
C4FA3		1:0.3	4.73	2.295
C5FA0	5%	1:0	4.60	2.290
C5FA1		1:0.1	4.65	2.331
CFA2		1:0.2	4.76	2.353
C5FA3		1:0.3	4.82	2.390
C6FA0	6%	1:0	4.93	2.362
C6FA1		1:0.1	5.02	2.401
C6FA2		1:0.2	5.13	2.436
C6FA3		1:0.3	5.46	2.473
C7FA0	7%	1:0	5.24	2.440
C7FA1		1:0.1	5.31	2.443
C7FA2		1:0.2	5.44	2.448
C7FA3		1:0.3	5.72	2.451

## Data Availability

The study did not report any data.

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
