# Peer review of "Effect of Fly Ash on Leaching Characteristics of Cement-Stabilized Macadam Base"

_materials, 2021, doi:10.3390/ma14205935_

Round 1

Reviewer 1 Report

The results obtained are well known to materials scientists in the field of research on the hydration and hardening of cement stone and were described by Taylor in the classic publication "Chemistry of Cement", to which, by the way, the authors do not have a reference.

The article contains information on the formation of calcium hydrosilicates both in the structure of the cement stone and on the surface of the aggregate, not supported by physicochemical research methods (possibly microstructure or IR spectral analysis)

Many of the assumptions expressed by the authors are not supported by studies of the structure of the cement stone modified by fly ash: porosity, the formation of calcium hydrosilicates, no physicochemical studies of the cement stone have been carried out after the washout of calcium ions from the structure of the cement stone.

The advantages of the article include the methodology developed by the authors for analyzing the degree of leaching of calcium ions from cement stone, but it does not provide the formation of a full-fledged article that could arouse interest in the scientific community dealing with structural changes in a cement matrix during its operation in conditions of high humidity.

Therefore, I believe that the article needs revision in order to ensure scientific novelty, substantiation of the results obtained by additional research.

Author Response

Dear Editors and Reviewer:  
With regard to the valuable comments made by the reviewers, we have responded to them item by item. The specific responses are listed in the annex(materials-1392450-Response-1.docx).

Reviewer 2 Report

Manuscript number: Materials-1392450

Title: The effect of fly ash on leaching characteristics of cement stabilized macadam base

General Comment

This paper deals with the study of the influence of cement dosage and Fly Ash dosage on the Ca leaching properties of cement stabilized macadam base. The topic is not so original, as the pozzolanic activity of Fly Ash is well known and widely studied for many years. A chemical, mineralogical and microstructural characterization of the ash is completely lacking. The type of material studied is for very specific applications in the building sector. English is often quite cryptic, it should be revised by a native English speaker. Thus, in my opinion the paper must be rejected and encourage resubmission on a more specific journal.

Detailed comments

The authors should use line numbering when submitting a paper;

Page 2 line 14 – What is one-dimensional calcium? Explain;

Page 2 line 15 – What kind of resistance?

Page 2 line 23 – “Fly ash is an active material of volcanic ash” explain;

Page 2 line 29 – What is RCC?

Materials and methods – How were chemical and mineralogical compositions obtained?

Page 3 – What is lithotripsy?

Figure 2 – What does this figure refer to?

Figure 3 and followings – Check figure formatting;

Page 7 – “(2) Effect of cement dosage on leaching rate of calcium ion” How was Ca leaching rate calculated?

Page 8 – “When the base material is not mixed with fly ash, the cement dosage of the specimen should be greater than 4%. When fly ash is mixed in the base material, 4% cement dosage should be taken as the lowest cement dosage.” Greater than 4% or 4% is the lowest means the same;

Author Response

Dear Editors and Reviewer:  
With regard to the valuable comments made by the reviewers, we have responded to them item by item. The specific responses are listed in the annex(materials-1392450-Response2).

Reviewer 3 Report

Comments

This paper investigated the effect of fly ash on leaching characteristics of cement stabilized macadam base. The outcome of the paper is interesting however, there are several aspects that need to be improved. The reviewer can only recommend for publication if the author satisfactorily address the following major comments in the revised version.

  1. The experimental setup photos should be provided for better understanding.
  2. The research questions and justification of selected parameters should be highlighted.
  3. Which test standards was considered in this study? How many replicate samples were tested in each category?
  4. The deterioration mechanism of the specimen should be discussed more clearly.
  5. The novelty of the study should be highlighted more clearly at the end of introduction section. How this study is different from the published study in literature?
  6. How the outcome of this study will benefit researchers and end users? This need to be highlighted in introduction or end of conclusion.
  7. The importance of fly ash research and the recent investigation in this area should be discussed in introduction section to improve the background study. Recently, the global generation of fly ash is presented in [Ref: Recycling of landfill wastes (tyres, plastics and glass) in construction – A review on global waste generation, performance, application and future opportunities], and its application is highlighted in [Ref: Investigation on the physical, mechanical and microstructural properties of epoxy polymer matrix with crumb rubber and short fibres for composite railway sleepers]. Suggest to include them in introduction section with proper citations to improve the background study.

I would be happy to see the revised version to understand how these comments are being addressed.

Author Response

Dear Editors and Reviewer:  
With regard to the valuable comments made by the reviewers, we have responded to them item by item. The specific responses are listed in the annex(materials-1392450-Response3).

Round 2

Reviewer 1 Report

Thanks for replies to my comments.

Reviewer 2 Report

The objectives of the study are clearer now and the authors correctly addressed my particular concerns. English was improved.

Reviewer 3 Report

I have no further comments